# Nerve Ultrasound in Traumatic and Iatrogenic Peripheral Nerve Injury

**DOI:** 10.3390/diagnostics11010030

**Published:** 2020-12-26

**Authors:** Juerd Wijntjes, Alexandra Borchert, Nens van Alfen

**Affiliations:** 1Department of Neurology and Clinical Neurophysiology, Donders Institute for Brain, Cognition and Behavior, Radboud University Medical Center, 6500 HB Nijmegen, The Netherlands; Juerd.wijntjes@radboudumc.nl; 2Nervenultraschall Euregio, Pauwelsklinik, 52064 Aachen, Germany; alexandra.borchert@gmx.de

**Keywords:** peripheral nerve, fascicle, nerve injury, nerve trauma, nerve ultrasound, ultrasonography, imaging, neuroma, iatrogenic, nerve surgery

## Abstract

Peripheral nerve injury is a potentially debilitating disorder that occurs in an estimated 2–3% of all patients with major trauma, in a similar percentage of medical procedures. The workup of these injuries has traditionally been clinical, combined with electrodiagnostic testing. However, this has limitations, especially in the acute phase of the trauma or lack of any recovery, when it is very important to determine nerve continuity and perform surgical exploration and repair in the case of the complete transection or intraneural fibrosis. Ultrasound can help in those situations. It is a versatile imaging technique with a high sensitivity of 93% for detecting focal nerve lesions. Ultrasound can assess the structural integrity of the nerve, neuroma formation and other surrounding abnormalities of bone or foreign bodies impeding the nerve. In addition, this can help to prevent iatrogenic nerve injury by marking the nerve before the procedure. This narrative review gives an overview of why and how nerve ultrasound can play a role in the detection, management and prevention of peripheral nerve injury.

## 1. Introduction

Peripheral nervous system (PNS) trauma is an infrequent but potentially debilitating disorder, that can have a profound impact on a patient’s wellbeing and functional abilities [1]. The overall incidence is estimated to occur in around 2–3% in patients presenting to major trauma centers globally [2,3,4,5,6], and amounts to about 13–23 per 100,000 persons per year [7]. Specific trauma mechanisms have a higher chance of injury to specific nerves, an example being the radial nerve that is affected to some degree in about 10% of the patients with a traumatic humeral shaft fracture. It is estimated that about 1.5–2% of the patients with a crush injury or joint dislocation will also have PNS injury. Knowledge of the specific trauma mechanism is mandatory to predict which nerve(-s) will be affected, and this knowledge should prevail over the standard neurologic methodology to localize the PNS disorder during the workup.

Clinically, a PNS injury should be suspected in any patient who has a trauma or surgical wound but also has:More pain than is commonly expected to occur from the traumatic lesion or medical procedure itself;Decreased function or weakness in muscles distal from the traumatic lesion or medical procedure;Paresthesias and/or hypesthesia in skin areas around or distal from the traumatic lesion or medical procedure;A sharp injury mechanism with the visible transection of other soft tissue structures (e.g., blood vessels, tendons).

When the trauma was a medical procedure, the disorder was called iatrogenic PNS injury. Although this may pose an uncomfortable topic for the treating physician and hospital insurers, it is well known that certain procedures carry a small but inherent risk of inadvertent injury to adjacent nerves, even when performed correctly [8,9]. A frequent example is a traction injury of the peroneal (fibular) nerve portion of the sciatic nerve during hip joint replacement surgery, this occurs in 2–5% of patients during the first procedure and in 10% or more during revision surgery [10]. Other well known risky procedures are cervical lymph node excision (3–6% risk to the accessory nerve) and humeral shaft osteosynthesis after fracture that carries an additional 5–15% risk of radial nerve injury on top of the risk posed by the fracture itself.

Traditionally nerves have been examined by electrodiagnostic testing, using nerve conduction studies and needle electromyography(EMG) [11,12]. However, electrodiagnostic tests are inherently uncomfortable and not well suited for the acute phase, as their maximum diagnostic information only becomes available after 2 weeks when Wallerian degeneration is complete. Moreover, electrodiagnostic testing cannot provide information on the exact localization of the lesion site in patients with a complete loss of nerve conduction. The past decade has seen the advent of nerve imaging, with ultrasound and MRI as additional tools to assess PNS integrity [13,14,15]. While MRI provides a wide field of view and can depict nerves irrespective of their depth or the interposition of bony structures (such as the intrapelvic lumbosacral plexus), ultrasound has a higher focal resolution for superficial nerves within 5 cm from the skin surface, which can be performed at the bedside, and has no contraindications. Additionally, in a head-to-head comparison, nerve ultrasound was better at detecting pathology with 93% sensitivity compared to the 67% sensitivity of MRI, and an equal specificity of 86% [16]. Ultrasound was found to modify the diagnosis and therapy in 58% of patients with traumatic nerve lesions when added to the standard neurophysiologic workup [14].

This review discusses the role that peripheral nerve ultrasound can play in the diagnosis and management of patients with (suspected) PNS trauma. We will focus on the clinical challenges that benefit from a targeted ultrasound investigation, such as scanning in the acute phase of trauma, the identification of the lesion site and nerve continuity, and the assessment of the anatomical relation of the nerve to its surroundings. This review will highlight the appearance of scars, adhesions and osteosynthesis material, neuroma outgrowth and remodeling, as well as describe the role of ultrasound in the workup for surgical intervention in nerve trauma, the role ultrasound “sono-” palpation and ultrasound guided injection can play in identifying the source of pain symptoms, and the potential role of ultrasound for iatrogenic injury prevention.

## 2. Introduction of Ultrasound Technique for Nerve (Trauma) Scanning

Normal nerves have a specific ultrasound aspect that is easily recognized from their gross histology appearance or from the direct inspection of a cut nerve during surgery. Figure 1 shows a normal transverse ultrasound image of a median nerve in the forearm with these comparisons. 

Peripheral nerves are quite similar to high tension electrical cables (for an example see https://en.wikipedia.org/wiki/High-voltage_cable), that are composed of several smaller cable bundles (the fascicles), each containing multiple copper strands (axons) of insulated (myelin sheath and Schwann cell) electrical wiring, contained in strong individual sheaths (perineurium) and bundled together by a protective outer sheath (epineurium). Ultrasound is well suited to display this cable-like array in real time and with great detail, with a lateral resolution of around 0.1–0.2 mm for the distal limb nerves using most modern ultrasound systems.

To create an ultrasound image, a high-frequency linear transducer in the 12–24 MHz range is used with a musculoskeletal ultrasound preset. For deep nerve segments, such as the sciatic nerve in the thigh and buttock region, it can be necessary to use a lower frequency (2–9 MHz range) probe with a convex surface to be able to get a view of the nerve (Figure 2). 

Plenty of ultrasound gel and a light touch during scanning should be used, especially in recent trauma patients. In vulnerable patients or when scanning in wound areas, using sterile gel and good antiseptic precautions is recommended, including disposable non-sterile gloves and cleaning the ultrasound machine and transducer prior to and following scanning. 

As a standard in neurological nerve ultrasound, transverse imaging is performed for the anatomical identification of the nerve of interest, and to assess its size and architecture for abnormalities. The international agreement is that the nerve size is best measured as a cross-sectional area (CSA) of the nerve traced within the hyperechoic outer epineurial rim [17]. Tracing within the epineurium is the best way to standardize the measurements across patients and over time, as the outer parts of the epineurium blend into surrounding epimysium and fascial structures, which hampers the accurate delineation of the nerve. CSA reference values are available for many nerves [18]. Nerve size increases during growth, so for schoolchildren it is advisable to adhere to a reference that is 50–75% of the adult size depending on age [19].

To assess the nerves for abnormality, it is strongly advised to scan the nerve all the way along its accessible length, looking for sudden changes in size or appearance. When an abnormality is found, it is advisable to also twist the probe 90° around and make a longitudinal image of the lesion site. When measuring neuroma sizes, placing several diameter markers proximal, at and caudal to the lesion site may be helpful (Figure 3).

Longitudinal images are not very good for the identification of the anatomical location, but they are more intuitive for looking at pathology by referring physicians (or patients). It is advisable to annotate images with sparse text to indicate the site, including a nerve name abbreviation such as “MED” for the median nerve, and an indicator for left/right, distal/proximal, etc.). For any nerve that could require surgical intervention, it is recommended to measure the site of abnormality in reference to a recognizable anatomical landmark (e.g., “5 cm distal from the intermalleolar line, 2 cm lateral from the midline” etc.). We strongly recommend making a short ultrasound video scanning from the proximal across a lesion site to a distal one, as ultrasound videos are much easier to interpret afterwards than still images. Finally, the images should be saved to a network location or printed to be saved with the patients’ health records.

For optimal images, it is paramount that the transducers’ ultrasound beam is directed at the nerve of interest at a 90° angle, to make sure that all sound that encounters the nerve will be reflected back to the probe surface for display on the screen. This seems straightforward but can be technically challenging for the examiner who must follow all the long limb nerves that can run in many different kinds of angles and directions below the skin and in between other tissues such as the fascia, muscles, blood vessels and bone. In some regions, for example the axilla, it will not always be possible to position the probe in a way so that the necessary 90° angle can be obtained, especially in patients with a limited range of motion due to trauma or pre-existent shoulder pathology.

Ultrasound cannot see through air, bone, or other objects with a completely different composition than body tissue, such as bandages and plaster casts, plastic wound foil with air underneath or drains, metal wiring, or osteosynthesis screws and plates (Figure 4).

While ultrasound has no inherent contra indications, good communication with the referring physician is needed to ensure that the nerve of interest is accessible for ultrasound. Fresh scars or small wounds can be scanned by covering them with a patch of transparent plastic film dressing with some NaCl or sterile gel below the film, making sure that no air bubbles remain underneath. It is also important to ensure the patient can be examined comfortably, with enough support of the affected limb and analgesia if needed.

Basic findings in nerve trauma include focal enlargements indicating neuroma in continuity, with or without the disorganization of the internal fascicular structure, or partial or complete transection of the nerve. Swelling from traction or compression injury already occurs in the first few hours following the injury. During recovery, such a neuroma in continuity may undergo remodeling with the normalization of the fascicular architecture and a decrease in size. However, in many cases, the nerve appearance will remain slightly altered, with a perceptible increase in both fascicular size and the thickness of the perineurial and epineurial rims. In the case of complete transection, both ends of the nerve can usually be found a few centimeters apart, as the elasticity of normal nerve tissue causes a recoil when the nerve is cut. Both transected nerve ends will form into stump neuromas within days, and if no re-approximation occurs surgically, will remain so indefinitely.

## 3. General Role of Ultrasound in Nerve Trauma

### 3.1. Identification of the Lesion Site

High-resolution ultrasound is an important tool in the detection and localization of traumatic nerve lesions. It can help to localize the lesion exactly where other modalities such as the clinical information, EMG results or MRI cannot [20]. This precise localization is important for two reasons: it will predict the expected recovery time of muscle and skin innervation and allows for targeted surgical intervention in situations where no clinical recovery is expected. Electrodiagnostic testing, including needle EMG and nerve conduction studies, is the standard method for assessing peripheral nerve injury, that can provide valuable prognostic information by showing the extent of denervation and reinnervation [12]. However, EMG has limitations: in the case of severe axonal damage or persistent conduction block, no action potential can be recorded. When no response is obtained, electrodiagnostic testing cannot differentiate between complete axonotmesis, neurotmesis (Seddon grade 5), or the different degrees of intraneural damage (Sunderland grade 2–4) [21,22]. Practically, some patient groups such as children often do not tolerate EMG. Furthermore, the lesion site can be a partial lesion in the proximal part of a nerve corresponding to the somatotopic fascicular organization of nerves, imposing clinically and electromyographically as a distal lesion [23].

The early detection of a lesion site and type is very important for prognosis and outcome because there is a limited timeframe for effective reinnervation, and in case this fails, the initiation of other therapeutic steps including nerve surgery. Nerve recovery is initiated right after axonal transection, when the denervated muscle fibers and skin areas start sending out neurotrophic signals that will attract any remaining axons in the vicinity and cause them to sprout to the denervated tissue. The quickest way for reinnervation is so-called collateral reinnervation, where the remaining axons in the fascicle take over the innervation of motor units and skin areas of their damaged neighbors. In case of severe injury, when more than approximately 75% of the axons in a fascicle are lost, the remaining axons in the bundle will not be able to reinnervate every motor unit and skin area, and reinnervation will have to come from the proximal ingrowth of new axons from the site of the lesion. Proximal reinnervation proceeds at about 1 mm/day. If the distance between the nerve lesion and target muscle is too large, for example, distal muscles of the lower leg and sciatic nerve injury, successfully reinnervation fails, because over time there are irreversible changes in the muscle and neural axon tubes that do not facilitate further outgrowth and reinnervation [24]. In sensory nerves, there seems to be more time for reinnervation. In addition, damage to the nerves’ connective tissue structures may lead to perineural fibrosis and scarring, which can prevent further axon outgrowth. 

The therapeutic window for nerve surgery depends on an accurate assessment of the extent of the damage, which can usually be done clinically and may be augmented by EMG, localizing the exact site of the lesion and hence the distance between the lesion site and the affected muscle and skin, and detecting morphologic alterations of the damaged nerve that indicated transection or intraneural scarring (Figure 5). This information will guide surgical decision making by assessing the regenerative potential [25].

In addition to morphologic aspects, the Tinel sign during ultrasound examination can help in identifying the site of a nerve lesion. Jules Tinel described in the early 1900s tingling with the light palpation of injured or regenerating nerves [26]. The compression of the ultrasound probe, sometimes dubbed “sonopalpation” can guide us to the lesion site by provoking the Tinel sign when the transducer moves over the affected nerve [27]. In our opinion, sonopalpation is also a good way to test whether a nerve lesion visualized by ultrasound is a likely cause of persisting symptoms such as nerve pain or paresthesia; if a lesion is seen but no symptoms can be evoked on compression or palpation, it is less likely that this part of the nerve is the cause of ongoing symptoms. 

Ultrasound is also able to assess the nerve dynamically, i.e., when moving the muscles and joint. Dynamic ultrasound examination can uncover the adhesions of a nerve to surrounding tissue (fascia, tendon sheaths or scar), and can show nerves moving over or in an abrasive environment such as bone fragments or protruding osteosynthesis material. 

### 3.2. Assessing Nerve Continuity

An important aspect for choosing the right therapeutic procedure and thus influencing the prognosis of a nerve injury, is to differentiate between patients with higher Sunderland grade intraneural damage or even the transection of a nerve versus a less severe lesion with preserved nerve continuity and good regenerative potential. To categorize a lesion as major or minor based on imaging (Figure 6), a few typical sonomorphologic signatures can be used, that include the fascicle swelling and hypoechoity of the nerve, the absence of a normal fascicular pattern, and the continuity assessment to detect partial or complete nerve severance with the presence of neuronal stump [28,29].

In pure neurapraxia, a Sunderland grade 1 and the least severe lesion, there are either no pathologic findings on ultrasound or just a mild swelling of the nerve with an intact fascicular pattern. A Sunderland grade 2 injury is slightly more severe with axonal damage, and also shows clearly enlarged cross-sectional areas of affected fascicles and nerves at the lesion site because of axonal swelling and edema. Higher grade damage like in patients with Sunderland grade 3 and 4 lesions shows a loss of the normal nerve architecture and echotexture with a disruption of the fascicular pattern and often sizeable hypoechogenic enlargement of the nerve. The more extensive the loss of the normal nerve architecture, the higher the chance that the nerve will develop intraneural fibrosis that hampers or prohibits recovery. For these lesions, a subdivision of the Sunderland system has been described by Millesi [30]. A Sunderland grade 5 lesion shows a transection of the nerve with a loss of nerve continuity. An overview of the nerve injury classification can be found here: https://en.wikipedia.org/wiki/Peripheral_nerve_injury_classification.

The early identification of these Sunderland grade 4 and 5 lesions is very important for predicting the outcome and the need for surgery [21,22]. High-resolution ultrasound allows this early categorization of lesions to guide surgical decision making [31]. Surgery is always indicated if nerve continuity is lost (i.e., Sunderland grade 5). In the case of a complete transection, the proximal and distal nerve stump will be separated, and as the regenerating axons in the fascicles have no guiderail for regrowth, they will form a stump neuroma as they sprout. A similar situation can occur if there is no gap, but so much intraneural damage that the nerve forms an internal “wall“ of fibrosis (i.e., Sunderland grade 4), through which axons cannot regrow (also see below). This will result in a large disorganized swelling at the lesion site, described as a neuroma in continuity.

In everyday clinical practice, there are patients with a nerve lesion that does not fit exactly one distinct category or classification. In these heterogenic lesions, different nerve segments and ultrasound images can show different grades of damage within one nerve. There can be intact fascicles next to fascicles with partial fibrosis, next to fascicles without any preserved texture that show a neuroma. Typical findings of such heterogenic lesions can be fusiform swelling of the nerve with a hyperechogenic texture, the destruction of the fascicular pattern but intact epineurium, the partial transection of a few fascicles, swelling and hypoechogenic texture with or without intact epineurium, and with or without signs of external compression of perineural tissue like callus or scar. It is very helpful for the further management and prognosis to describe the ultrasound findings exactly and specify the lesion type in its heterogeneity instead of diagnosing a “partial nerve lesion“. The potentially heterogenic affection of a damaged nerve is also why knowledge of the trauma mechanism, clinical and EMG findings by themselves do not necessarily or automatically lead to the right therapy. In the case of a nerve traction injury, for example, the ultrasound can show undulating fascicles, elongations or even loop formation, with different regenerative potential for the different lesion sites. The focal injury of the perineurium can induce fascicular herniation into the nerve, also with an adverse effect on prognosis [32]. In such cases, high-resolution ultrasound can provide the needed detail to make the appropriate therapy choice [31].

## 4. Ultrasound in Different Nerve Trauma Mechanisms: Compression, Traction, Crush, Thermal, and Transection Injury

### 4.1. Compression Injury

Compression injuries can either be acute or chronic. Acute compression injury, such as a peri-operative ulnar compression neuropathy, typically presents with transient paresthesia, numbness and accompanying weakness that recover in weeks. Chronic compression, e.g., carpal tunnel syndrome, is often a progressively worsening condition that persists without proper intervention. Clinically, symptoms start with paresthesia and often also pain, and progress over time to hypesthesia and muscle weakness, depending on the extent of axonal damage [33]. In the case of nerve entrapment, a focal area of nerve enlargement can be seen just proximal to the site of compression. Ultrasound not only helps with localizing the compression site but could also help in identifying its cause. An example of such nerve entrapment, caused by osteosynthesis material in this patient, is shown in Figure 7B. In this case, the radial nerve shows the focal enlargement just proximal to the screw where it is compressed. In addition to a CSA increase, this caliber change can be appreciated quite well in the longitudinal images. When entrapment is severe or chronic, an associated loss of the internal fascicular architecture can be observed as well. 

### 4.2. Traction Injury

Traction injury can occur during trauma, for example, in obstetric or adult plexus brachialis injury, or during interventional procedures, such as nerve injury caused by a skin retractor, or the bending open of a joint during replacement, such as during hip replacement surgery that can damage the sciatic nerve. Depending on the how far and how long the nerve was stretched, different degrees of neural damage can occur. Acute stretch will cause the nerve trunks to first undergo compression with increasing stiffness during traction [34]. The fascicles are the main stress-bearing element of the nerve. Nerve fibers (i.e., axons) will be damaged before there are signs of macroscopic nerve injury, and nerve ischemia will already occur at 8–15% strain, with vessel occlusion in strains > 15% [35]. The breaking strain of human nerves was experimentally determined in early cadaver studies and was shown to be from 50 pounds (i.e., 23 kg) in proximal nerves such as the brachial plexus to a minimum of 80 pounds (i.e., 36 kg) in distal arm and leg nerves [35]. A cause of this difference is very likely the increasing connective tissue content in nerves (which also translates to their increasing echogenicity), from 25% in the roots to 50–70% in the distal arm and leg nerves. However, subsequent research found large regional differences in the strain a particular nerve could undergo before being damaged, and showed that nerves across joints (i.e., the elbow, carpal tunnel, hip, ankle) can take much more strain than nerve segments in non-joint regions, that are likely adapted to the periodic strain caused by limb excursions [35]. In the non-joint regions, the nerve will likely face total mechanical failure at >30% stretch [36]. 

The initial therapy after traction injury is conservative and includes pain management, rehabilitative measures and a minimum follow up after 3 and 5 months. If conservative treatment fails, surgical exploration and reconstruction should be considered within 6 months [37,38]. Depending on the severity of the injury, the nerve will appear hypoechoic and swollen, with or without a disruption of the fascicular architecture. Figure 7D shows an example of a traction injury of the radial nerve, just proximal from a humerus fracture site. In the longitudinal images, the epineurium of the nerve can be seen to be intact. 

A particular feature of traction injuries, that one has to be aware of, is the mechanical effect that can occur on more distal parts of the nerve, away from the site where it was primarily damaged. The most vulnerable distal nerve regions are the points where the nerve is anatomically anchored to its surroundings, such as at the level of a bifurcation, or beneath or between a strong fascial structure. So, in sciatic nerve trauma one has to actively look for a second lesion in the common peroneal nerve at the level of the fibular head, that might improve with surgical decompression. A similar phenomenon can be observed in axillary plexus trauma with the secondary affection of the musculocutaneous nerve where it bifurcates between the biceps and brachialis muscle, and with the trauma of the radial nerve that can show secondary changes to the posterior interosseus nerve at the level where it enters the supinator muscle; although the latter does usually not need specific treatment.

### 4.3. Crush Injury

Crush injuries occur from the traumatic compression of the nerve by blunt force, such as from the nerve and surrounding tissue being hit with a bat or ran over by car, getting caught in industrial machinery such as crushing rollers, or pinched between surgical clamps. Crush injuries can cause different degrees of nerve damage ranging all the way from neurapraxia to (partial) neurotmesis. Complete traumatic disruption with nerve transection (i.e., Sunderland grade 5) is rare, but the internal nerve structure can be badly damaged (i.e., Sunderland grade 4). Recovery follows the rules of the Sunderland grading system. Nerve ultrasound at the site of the injury will show a hypoechoic and enlarged nerve with more or less disrupted fascicle architecture and in the acute phase ecchymoses and edema of the surrounding tissues, as shown for example in Figure 7C.

Of note, patients with crush injuries of the limbs have a significant chance of developing a compartment syndrome. It is therefore advised to surgically decompress the known entrapment sites in these limb regions (e.g., the carpal tunnel, ulnar tunnel, peroneal nerve around the fibular head and the tarsal tunnel) during the acute surgical management of these lesions, to prevent acute entrapment injury caused by the increased compartment pressure.

### 4.4. Thermal Injury

Burn injury causes peripheral nerve damage in about 6.4% of the patients, usually involving two or more nerves [39]. In severe third-degree thermal (heat) burns, peripheral nerves located directly under the skin and subcutis may be damaged either directly, by increased compartment pressure due to tissue edema, or late by the scarring of the burned area. The extent will depend on the depth of the burn injury. In second-degree burns, there is usually no involvement of the major limb nerves, but small skin branches may be affected by the burn itself or by scarring. The temperature and the length of the heat exposure will determine the extent of local tissue responses, such as the heat coagulation of proteins and the subsequent release of local mediators that cause edema and hyperemia. Figure 7F shows an example of the thermal injury of the ulnar nerve at the level of the medial epicondyle, just next to some osteosynthesis material. In this case, a hot K-wire was the cause of the thermal injury of the nerve. In the longitudinal direction, the interruption of the normal structure of the nerve by scarred tissue can be recognized. In large burns, a systemic response with inflammation and decreased tissue perfusion, electrolyte changes and the disruption of the microcirculation due to inflammatory effects may cause further nerve damage. 

Electrical burns are a special type of thermal injury, that can be caused by for example lightning strikes, work accidents with high tension equipment, or the failure of electrical safety in medical procedures or devices such as electrocauterizers. The incidence of nerve injury in electrical burns is estimated around 13.5% [40]. Electricity causes damage by the conversion of electrical injury to heat, where the amount of heat production is determined by Ohm’s law and the Joule effect of resistive heating. In addition, both nerves and blood vessels make preferential conduits for electrical current, predisposing them to damage, not just at the entry point of the current but also distant from it. Generally, high-voltage injuries (generally > 1000 V) can create irreversible damage. In low-voltage injuries (< than 1000 V) the degree of neural injury varies. Early surgical debridement of the burned tissues is often needed to limit the amount of further nerve damage. Nerve reconstruction after thermal injury using nerve grafts is challenging due to the commonly large defect area, the limited availability of donor nerves and most importantly, the poor quality of the wound bed. After nerve reconstruction, free tissue transfers are often needed to assure the good quality of the surrounding soft tissue [41]. The potential use of ultrasound in screening for suitable skin grafts was recently described [42].

Chemical burns are a third type of burn injury, caused by the molecular properties of an agent (e.g., acidic, caustic, oxidative, etc.) that reacts with human tissue. The extravasation of intravenous fluids and the spill of bone cement used for anchoring prosthetic components are known examples of chemical burn injuries in patients. Tissue contractures and scarring are the major complications of chemical burns. Just as in other burns, the thermal reaction can create a hyperechoic and swollen nerve, that may develop internal scarring (i.e., Sunderland grade 4) over time. 

### 4.5. Transection (“Sharp”) Injury

Complete nerve transection injuries are usually caused by “sharp” accidents, such a knife laceration, glass shard injury, bone drill or gunshot wound. Gunshot wounds, however, are complicated, as they can both damage the nerve directly, or damage the nerve with bone fragments from splintering upon impact, but can also cause severe traction/compression injury from the pressure waves that build up in the bullet trajectory or the subsequent compartment syndrome these pressure changes cause. After complete nerve transection (i.e., Sunderland grade 5, neurotmesis), the distal segment of the nerve will undergo Wallerian degeneration, and there is no chance of recovery across the gap as the proximal and distal nerve ends typically retract. In transection injuries, the nerve ultrasound can be used in the acute setting to reveal the discontinuation of the nerve, often best appreciated in the longitudinal direction. The hallmark is an interruption of the epineurial sheath and the fascicles. In transverse images, the nerve and fascicles in the stumps appear enlarged and hypoechoic, and over time this architecture will become more disrupted as stump neuromas develop (Figure 7E). Surgical treatment, with neurolysis and repair or grafting is the only option for nerve recovery; if this is not possible, tendon transfers and joint stabilization by splinting or arthrodesis should be considered.

## 5. Around the Nerve: Scars, Adhesions, Foreign Bodies

During the ultrasound scanning of nerve trauma patients, it is not only important to look at the nerve itself, but also to scan its anatomical surroundings for any abnormality that may impact on nerve recovery or pose an additional problem for the patient. The most commonly observed abnormality in trauma is scarring. As scars often run perpendicular to the skin surface, they are hard to visualize well with ultrasound, as the soundwaves travel parallel to the lines of the scar and almost nothing gets reflected back to the probe. Scars therefore show up as vaguely defined hypoechogenic longitudinal lines in the skin, subcutaneous tissue and fascia (Figure 8A), sometimes only visible by the interruption of other architectural features in those layers (Figure 8B). 

Scarring can also involve the nerve and its connective tissue ensheathing itself (Figure 9A), and can lead to a loss of nerve elasticity and adhesions to surrounding tissues (Figure 9B).

Scars also indicate the direction of tissue interruption and whether that may have impacted the nerve or not. They also show possible areas of tissue retraction in surgical cases of iatrogenic nerve injury. Traction on wound edges is an often overlooked but not infrequent reason for iatrogenic nerve damage and is relatively common in certain procedures such as lymph node excision in the posterior cervical region with damage to the accessory nerve mentioned earlier, and the surgical decompression of anterior tibial compartment syndrome with damage of the superficial peroneal nerve (Figure 10).

In recent wounds, stitches, surgical staples and drains may be still in place, that are usually echodense and scatter soundwaves from the surface, thereby limiting visual access to the underlying nerve (Figure 11). 

Ultrasound of fresh wound areas may be complicated by hematoma, giving a diffuse grayish shadowing of the image (Figure 12), and a loss of recognizable anatomical features (Figure 11 B).

Another common finding is the presence of osteosynthesis material in the vicinity of the injured nerve. Fixation plates and screws and K-wires can usually be visualized with ultrasound as hyperechogenic edges and structures that cover or pierce a bony cortex in the vicinity of the nerve (Figure 7F). It is well known that the placement of osteosynthesis material for certain indications poses a risk to peripheral nerves in that area, such as the radial nerve during humeral shaft fixation, the suprascapular nerve during metaglenoid fixation in reversed total shoulder arthroplasty, and the ulnar nerve during temporary elbow fracture fixation (Figure 13).

Sometimes it is not the osteosynthesis or prosthetic material itself that poses the risk, but the approach the surgeon must take to get the material in place. Hip joint replacement surgery is notorious for its risk of traction damage to the fibular part of the sciatic nerve, when the hip joint has to be bent open to insert the prosthetic shaft and ball and the cup part of the prosthesis [10]. As the fibular part of the nerve is fixed by the biceps femoris tendon insertion in the proximal part of the lower leg, traction on the sciatic nerve produces the greatest tensile stress on this part of the nerve, leading to injury in about 5% of patients having their primary surgery.

## 6. Ultrasound in the Acute Phase of Trauma

In the acute setting, the diagnosis of traumatic nerve injury can be challenging. The focus and efforts in the prehospital and emergency room phase are usually targeted at stabilizing the patients’ airway, breathing and circulation, and nerve injury can easily be overlooked. Often this will only become evident when the patient is awake on the ward. In the first 2 weeks, EMG cannot be used reliably to assess the extent and severity of the lesion, because Wallerian degeneration must be awaited before the denervation that follows from axonal damage can be observed [43]. In this acute phase, peripheral nerve imaging with ultrasound or MRI can fill the “electrodiagnostic time gap” by directly showing nerve (in-)continuity, directing appropriate treatment [44]. We recommend nerve ultrasound for all patients in the acute phase of peripheral nerve trauma, especially when there is a lack of clinical proven nerve-continuity. If the ultrasound cannot demonstrate certain nerve continuity, additional investigations such as surgical exploration, MRI and EMG need to be performed. A very specific case where additional imaging is always needed are patients in the acute phase of brachial plexus trauma with clinical signs of complete sensory and motor function loss, where MRI should be added to ultrasound to assess the integrity of the intraspinal nerve rootlets if there is no clear root avulsion extraforaminal on ultrasound.

Although MRI by itself is a widely available technique, specific MR neurography protocols are not, and high-resolution nerve imaging with MR is currently even less common in practice than ultrasonographic nerve imaging. In addition, it can be a daunting challenge to get the patient to the MRI in the acute phase setting of multitrauma, when the patient is intubated on the intensive care unit. Moreover, non-MRI compatible osteosynthesis material or osteosynthesis material in the vicinity of the suspected nerve lesion (such as an internal humerus bone pen in case of suspected radial nerve injury) can hamper adequate MR scanning. In all these examples, nerve ultrasound provides a good alternative for assessing the nerve immediately and at the bedside after a trauma.

## 7. Neuroma Outgrowth and Remodeling during Recovery

Depending on the extent of the connective tissue trauma, nerve recovery will give rise to changes in the shape and size of the nerve under study. A traumatic neuroma represents a disordered, hyperplastic growth response to nerve injury, that can give an increase in nerve size from 125 to 1600% of its normal size in the first 1–2 months following the injury [45] (Figure 14). 

In case of a nerve injury of Sunderland grade 3 or 4, the disruption of the endoneurium and next the perineurium will lead to the formation of intraneural fibrosis during recovery. This intrafascicular fibrosis can delay, divert or block axonal outgrowth, and in addition lead to the constriction of nerve segment by scar tissue [46]. These mechanisms lead to a disorderly growth of axons that result in a fusiform swelling of the injury site. Many of these axons will not reach their original target tissue (i.e., misrouting), and some will be unable to reinnervate any muscle or skin at all. In the case of full nerve transection, or Sunderland grade 5 lesion, the severed nerve ends will retract over the length of a few centimeters because of the epineurium’s elastic properties. The stumps will also form neuromas, while the distal nerve segment fasciculi will undergo a contraction of the endoneurial tubes after Wallerian degeneration has occurred, so that the cross-sectional fascicular area can be reduced up to around 60% after 3 months [46]. No direct relation seems to be present between the neuroma size and nerve function or recovery, although patients with very large neuromas of more than 5x the normal nerve size usually have no function left in the affected nerve and do not improve [45]. In patients that show clinical recovery over a period of 3–9 months, the neuroma will usually undergo remodeling and decrease in size, but possibly not all the way back to its healthy state. This slight residual enlargement following (any type of) nerve pathology, e.g., also inflammatory or ischemic lesion besides trauma, has been dubbed a "damage pattern" by some.

## 8. Ultrasound in the Prognosis and Workup for Surgical Intervention in Nerve Trauma 

Surgery plays a role in both the acute and chronic phase of nerve trauma. In the acute phase, surgical exploration is needed for all patients with sharp injuries who have damage to their tendons or blood vessels, as the chances of concurrent nerve injury are very high. Surgical exploration is also needed for closed injuries with a clinically complete nerve lesion and a high a priori chance of disruption based on the injury mechanism (e.g., root avulsion in brachial plexus injury) in whom imaging is not possible but surgical intervention could improve prognosis (but not in for example complete lumbosacral plexus lesions with no chance of distal nerve recovery because of the distance). In these cases, it is not mandatory to perform or wait for imaging results.

In the more chronic cases of nerve trauma, several factors must be considered when considering surgical treatment: the lesion site, the degree of nerve damage (combining clinical, electrodiagnostic and imaging information), the subsequent regeneration potential and the timing of the intervention. The assessment of the degree of nerve injury and the lesion site using ultrasound has been discussed above. Clinically, the injury degree and reinnervation potential can be inferred from the rate of symptom improvement, but this takes time. In the acute situation, it is usually not possible to say for sure that no recovery can occur, unless the nerve has been transected, and either cannot be repaired or has a lesion so proximal that no reinnervation of the target muscle and skin can be attained within 1–1.5 years. Even in milder lesions, a complete nerve conduction block may give rise to clinical paralysis with anesthesia, but this can recover within days to weeks. If the patient reports significant improvement within 4–6 weeks, the final prognosis, regardless of electrodiagnostic or imaging results, is usually good. If, however, no improvement is seen within 3–4 months, a severe axonal lesion can be inferred with insufficient remaining axons for complete collateral reinnervation. This is when the question of surgical treatment comes into view. 

So, the clinician has to decide if there is still time to wait and see, and for how long. With the help of ultrasound to locate the lesion exactly, an estimation can be made of when proximal reinnervation is to be expected, based on the distance the nerve has to regrow. Proximal reinnervation proceeds at about 1 mm/day in the limb nerves, and a bit faster at 2–3 mm/day in the proximal root and plexus segments [25]. Age is also an important factor, as elderly people (> 60 years) have a much smaller chance of nerve recovery than children or young adults. On ultrasound, the number of intact fascicles seems to be positively correlated to the chance of recovery [47]. Nerve edema will usually decrease with successful reinnervation. 

In patients that show clinical improvement over time but have a neuroma in continuity on ultrasound, one can wait and see, if a regular follow-up every 4–8 weeks is ensured to see if recovery is progressing as expected [25]. If there is doubt about the recovery, or some parts of the nerve recover but other fascicles do not, then surgical intervention is considered. Depending on the constellation, surgeons have several options to improve nerve recovery: neurolysis, grafting, split nerve grafting and nerve transfer [48]. After a complete transection, a nerve interponate is the only option for nerve recovery (Figure 15).

Another important clinical situation with an impact on scheduling the operation is the occurrence of a complete nerve disruption (i.e., Sunderland grade 5) after blunt force injury. Here, early surgical reconstruction is not recommended, because the blunt force impact can stretch well beyond the rupture site, and neuroma formation will take place in the injured nerve stumps when direct coaptation is attempted. Here, the outcome improves if surgery is performed after the neuromas are formed and can be clinically demarcated from healthy nerve during surgery. Ultrasound can track this neuroma formation and identify when surgery becomes useful, often only after 3 months or more have passed. Often this type of intervention will require the use of a nerve conduit interponate for tension-free coaptation.

On the other hand, there are clinical situations when a "wait and see" approach is not useful, because this would worsen the potential outcome. This would apply to, for example, proximal lesions where the reinnervation distance between the lesion site and first muscle is too long to wait. In addition, surgery should also not be delayed when the injured nerve shows persistent entrapment by osteosynthesis screws or plates, or bone fragments, scars or an extensive hematoma or seroma causing pressure. The ultrasonographer should actively look for these problems, especially in sites where the nerve is enlarged or undulating in its course.

## 9. The Role of Ultrasound in Chronic Pain Following Nerve Trauma

An estimated 10% of patients develop an incapacitating painful neuroma following PNS injury [7], either in continuity or as a stump. Ultrasound imaging can detect those neuromas, even if some are very small (e.g., distal sensory nerve neuromas with a CSA of 2 mm^2^ where the normal nerve size is 1 mm^2^). This can benefit the patient by diagnostic confirmation there actually is a nerve lesion (sometimes disputed in patients with ill-understood pain syndromes), and target the surgeon to the site of possible intervention, such as neuroma resection and the burial of the stump in muscle to decrease pain and minimize the chance of again developing a painful neuroma. This type of surgery will benefit an estimated 40% of the patients thus affected [49,50].

Ultrasound can also help the clinician test whether the neuroma is the actual or the main cause of the patients’ pain. In practice, this is done by either manipulating the neuroma with the probe or fingers under direct ultrasound guidance, asking the patient if that manipulation reproduces their typical pain sensation. Additionally, nerve ultrasound can guide the injection of a local anesthetic around the neuroma or slightly proximal to it in the main nerve, to temporarily block nerve function, and observe if that makes the typical pain go away and to what extent.

## 10. Preventing Iatrogenic Nerve Injury: Preoperative Use of Ultrasound for Injury Prevention

Many sensory peripheral nerves run in the skin and subcutaneous areas that form a surgical approach route to the operation. These nerves are therefore at risk of iatrogenic injury. Well known examples are the patellar branches of the saphenous nerve during knee surgery, the abdominal intercostal nerves during laparotomies, the superficial peroneal nerve during anterior tibial compartment syndrome release, the sural nerve during lateral ankle surgery, and the accessory nerve during lymph node biopsy. The course of these nerves makes them vulnerable to direct damage during incision, traction injury from skin and subcutis retractors, and stretch by the manipulation or compression of the joint to gain surgical access, or during the surgery itself with the use of clamps or osteosynthesis material [8,9]. 

Iatrogenic nerve injury is now considered a calculated risk in many procedures, that the patient should be informed about before the surgery. With correct anatomical knowledge of the anatomical course of the nerve, the surgeon can reduce the risk of damaging it. However, the exact course of these superficial nerve branches and the level of bifurcations into their terminal branches can be highly variable due to anatomical variation [51]. Nerve ultrasound offers the opportunity to evaluate and annotate the exact anatomic course of the nerve preoperatively. Some surgeons already use this preoperative nerve ultrasound imaging to minimize the tissue dissection and operating time, but somewhat ironically this is mostly done for the treatment of neuroma following iatrogenic nerve injury. [52,53]. There is not yet much information on the use of preoperative ultrasound to prevent nerve injury, but a few papers illustrate the useful role nerve ultrasounds can play in preventing iatrogenic injury [54,55]. Nerves at risk for iatrogenic injury that can be easily located and marked as pre-operative with nerve ultrasound, are summarized in Table 1 [9,56,57,58]. We would heartily recommend the use of ultrasound during the presurgical planning to reduce the incidence of iatrogenic nerve injury, as this would significantly reduce the burden of this infrequent but debilitating peri-operative complication in terms of patients’ distress, disability and litigations [9]. 

## 11. Conclusions

This review has provided an illustrated summary of the multiple roles nerve ultrasound can play in better diagnosing and guiding the management of patients with PNS injury. We hope it will help more clinicians incorporate this useful addition to their toolbox in everyday practice.

## Figures and Tables

**Figure 1 diagnostics-11-00030-f001:**
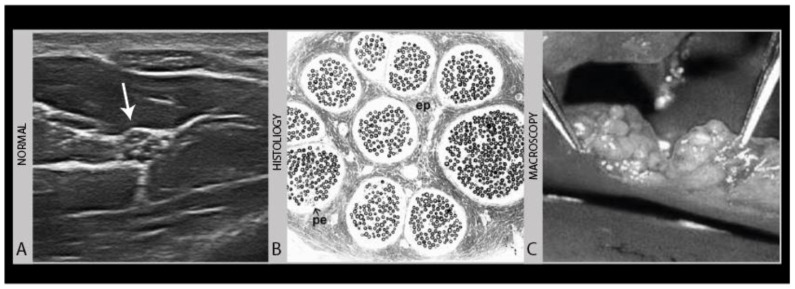
Normal transverse image of the median nerve mid-forearm (**A**), in comparison to a gross histology image (**B**) and the surgical appearance of a cut nerve (**C**).

**Figure 2 diagnostics-11-00030-f002:**
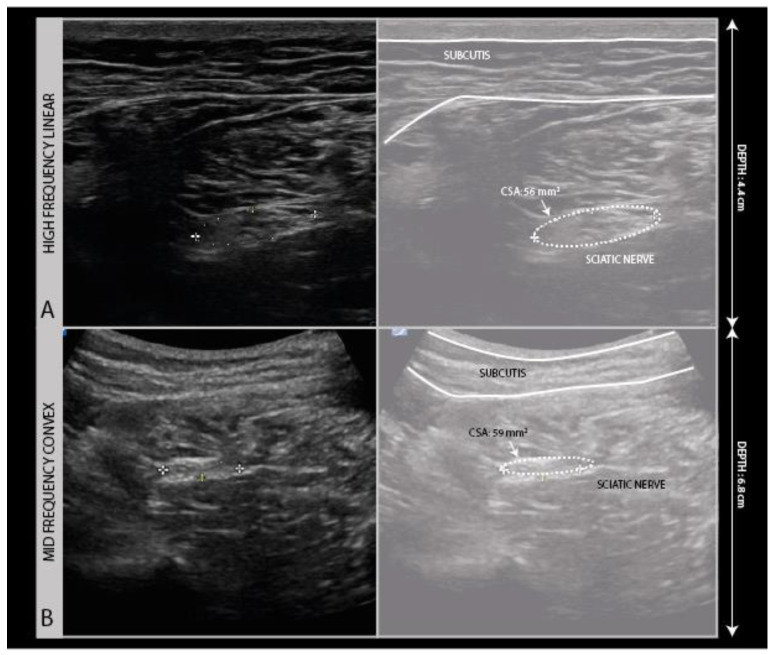
High frequency (5–16 MHz) probe image (**A**) compared to the low frequency (2–9 MHz) convex probe image (**B**) of a sciatic nerve traction injury following hip replacement surgery.

**Figure 3 diagnostics-11-00030-f003:**
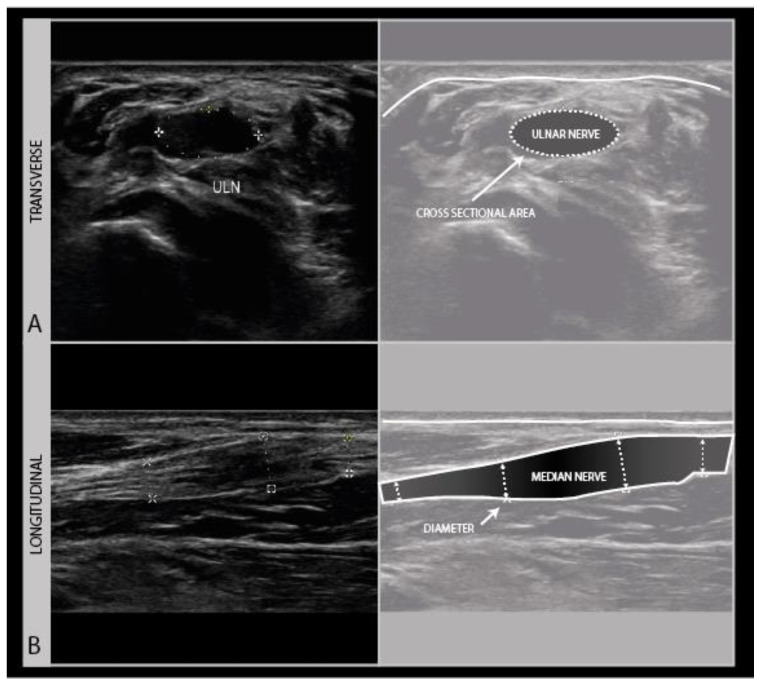
Image examples of transverse cross-sectional area (CSA) (**A**) versus longitudinal diameter measurement (**B**). A: distal ulnar neuroma; B: distal median nerve neuroma (adult male with a crush injury of the distal forearm). ULN = ulnar nerve.

**Figure 4 diagnostics-11-00030-f004:**
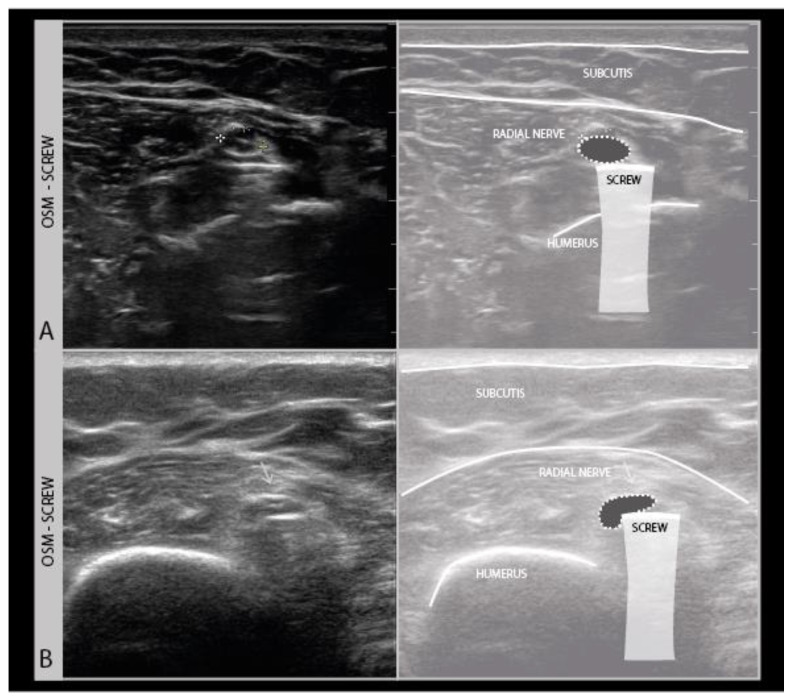
Two different examples (**A**,**B**) of the radial nerve crossing over a mid-humerus osteosynthesis material (OSM) screw (high frequency −18 MHz).

**Figure 5 diagnostics-11-00030-f005:**
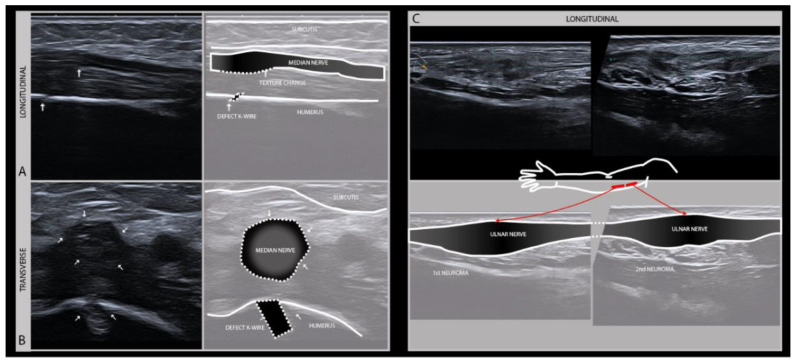
Longitudinal (**A**) and transverse (**B**) images of the left median nerve in the distal upper arm of a 5-year-old boy with a supracondylar humerus fracture surgically treated with K-wire fixation. The subsequent median nerve lesion with a change of nerve architecture seen near the bone defect after K-wire removal (high frequency probe 19–24 MHz). Full recovery within weeks after K-wire removal. Longitudinal (**C**) images of the ulnar nerve in the upper arm with two serial neuromas after a local gunshot wound.

**Figure 6 diagnostics-11-00030-f006:**
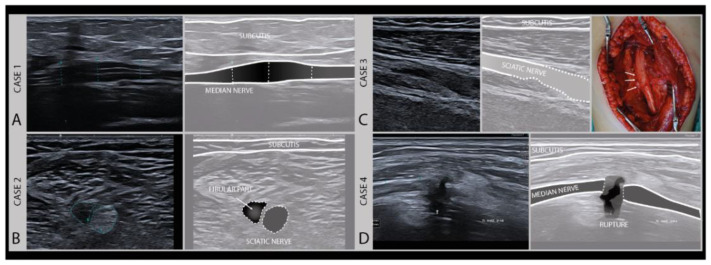
Examples of different degrees of axonal damage. Case 1 (**A**): median nerve swelling following local trauma (fracture), fascicles intact, full recovery without surgery (Sunderland II). Case 2 (**B**): heterogeneous lesion of the sciatic nerve (peroneal part) of a 19-year-old boy after a scooter accident. Case 3 (**C**): sciatic nerve damage (Sunderland III) in a 9-year-old girl after a severe car-accident (high-frequency 9–18 MHz). (Intraoperative image kindly provided by PD. Dr. F. Lassner). Case 4 (**D**): iatrogenic median nerve lesion with loss of nerve continuity (Sunderland V) in a 53-year-old man after arthroscopy of the elbow joint (high frequency 19–24 MHz).

**Figure 7 diagnostics-11-00030-f007:**
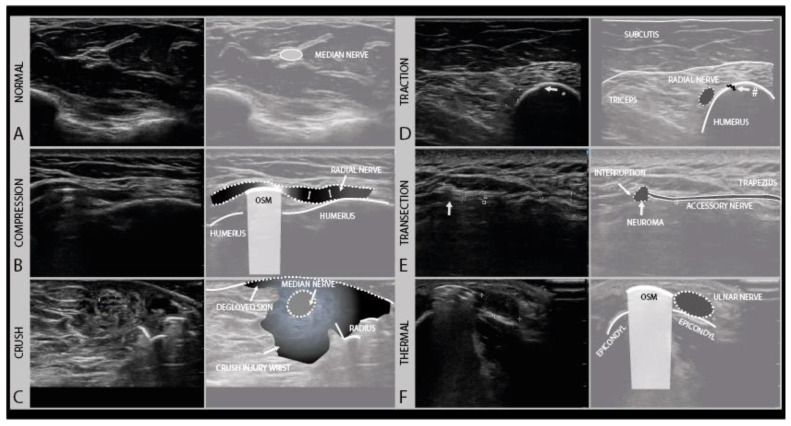
Examples of different trauma types. The normal transverse aspect of a median nerve in the forearm (**A**). Local compression injury of the radial nerve by local osteosynthesis material (OSM). Focal enlargement of the nerve diameter proximal to the lesion site can be seen (**B**). Crush injury of the median nerve at the level of the forearm with surrounding crushed subcutaneous tissue and degloved skin (**C**). Traction injury of the radial nerve after humeral fracture (#). Swollen and hypoechoic aspect of the radial nerve (**D**). Iatrogenic transection injury of the accessory nerve after resection of a local lipoma. A stump neuroma can be seen at the site of nerve injury, with a loss of continuity. (**E**). Swollen and hypoechoic ulnar nerve after thermal injury (hot K-wire) at the level of the medial epicondyle with in situ osteosynthesis material (**F**).

**Figure 8 diagnostics-11-00030-f008:**
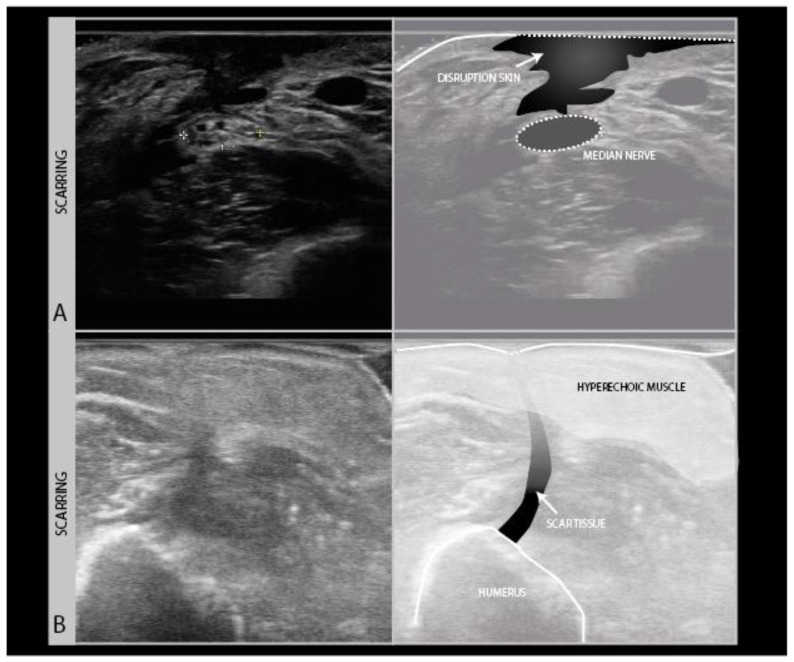
Examples of scarring of the skin and subcutis. Severe disruption of skin and subcutaneous tissue showing as a craggy hypoechogenic region over a median nerve neuroma in continuity in a patient with the piercing trauma of the forearm by a wooden fence (**A**). Hyperechogenic subcutis and muscle following surgery for an open humerus fracture, with a subtle vertical scar line interrupting the tissue layers (**B**).

**Figure 9 diagnostics-11-00030-f009:**
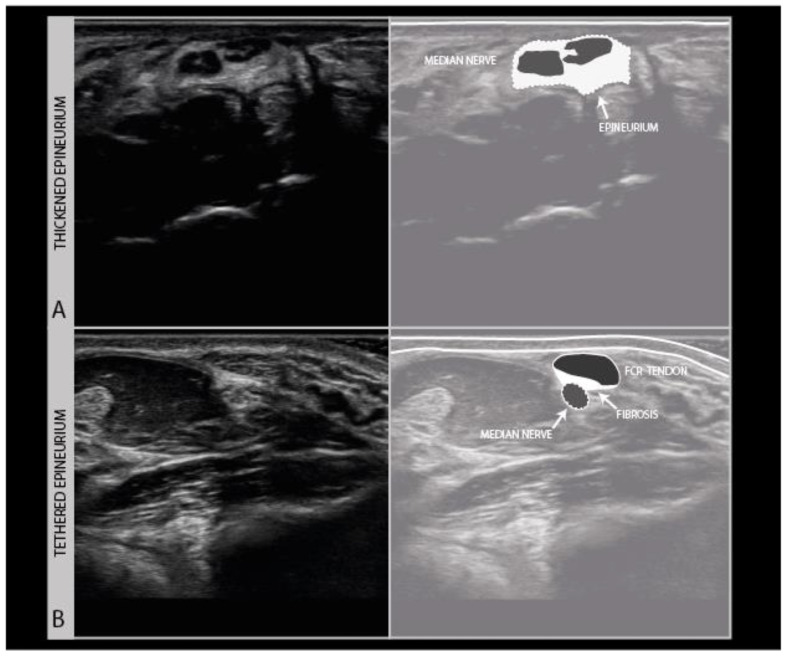
Persistent median neuropathy with very thick epineurial rim and hypoechogenic, swollen nerve fascicles, in a female patient who underwent 30+ carpal tunnel release surgical procedures (**A**). Median nerve adhesion in the distal forearm; the superficial epineurium of the nerve is tethered to the flexor carpi radialis tendon sheath directly on top of the nerve (hyperechoic region between the tendon and nerve) (**B**).

**Figure 10 diagnostics-11-00030-f010:**
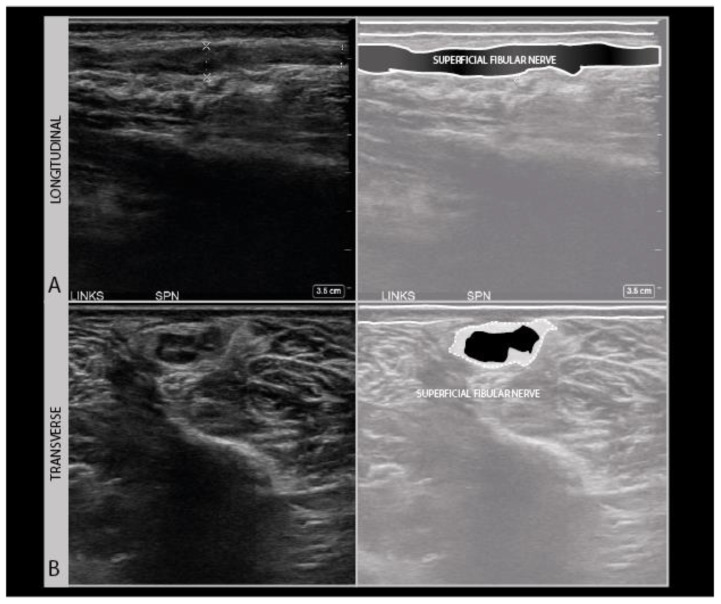
Disorganized and hypoechogenic superficial peroneal nerve neuroma in the distal lower leg in longitudinal (**A**) and transverse (**B**) direction; traction injury following surgical decompression with fasciotomy for anterior tibial compartment syndrome.

**Figure 11 diagnostics-11-00030-f011:**
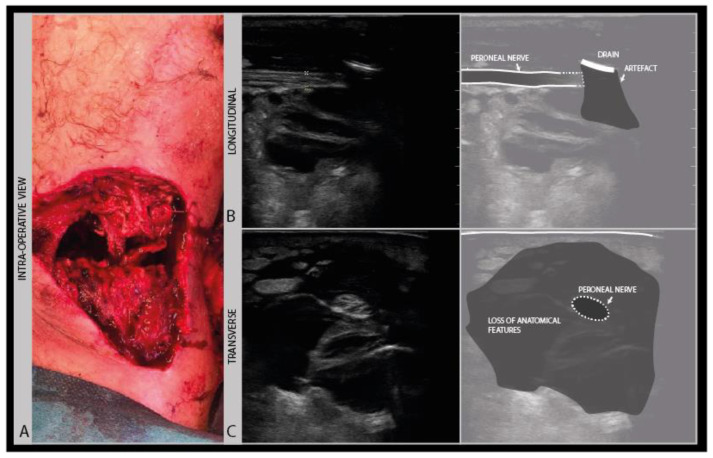
Severe knee torsion injury with a large tissue defect (**A**): post-surgical debridement ultrasound image of the posterior knee, with a tissue drain artefact obscuring the view of the underlying peroneal nerve (**B**) and a loss of anatomical features (**C**).

**Figure 12 diagnostics-11-00030-f012:**
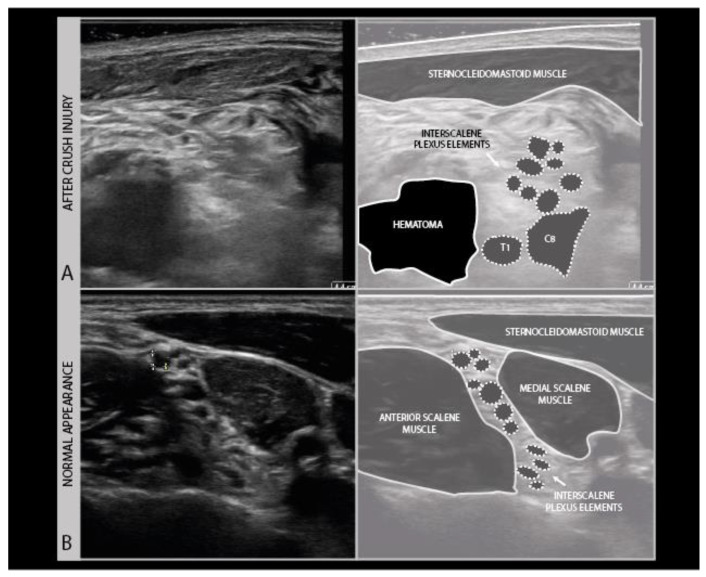
Right brachial plexus trauma with costoclavicular crush injury on day 1: hematoma with swollen C6–C7–C8–T1 elements (**A**) and normal interscalene plexus (**B**) for comparison.

**Figure 13 diagnostics-11-00030-f013:**
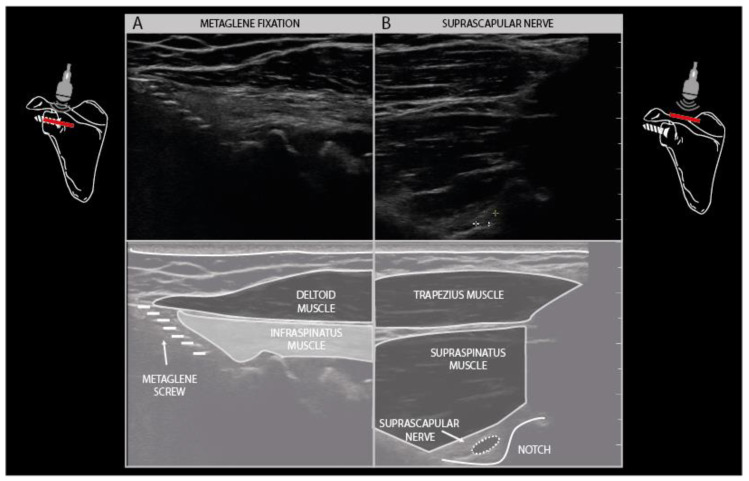
Reverse shoulder arthroplasty with metaglene fixation screw traction injury to the suprascapular nerve. Hyperechogenic, denervated infraspinatus muscle layer underneath a normal appearing deltoid muscle (**A**). Suprascapular neuroma in continuity (3× normal size) at the level of the spinoglenoid notch (**B**).

**Figure 14 diagnostics-11-00030-f014:**
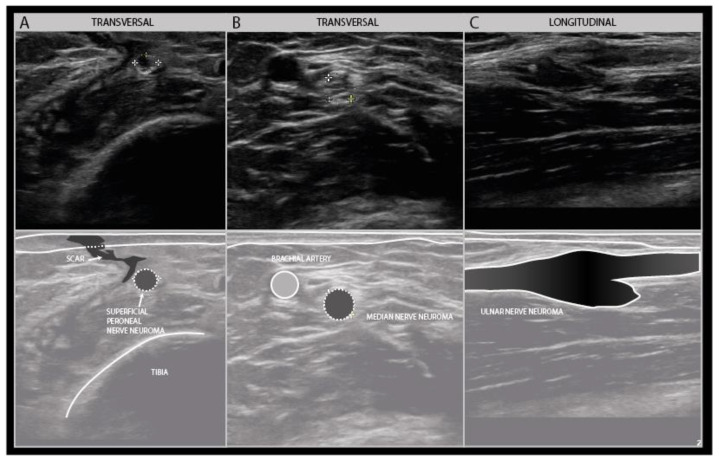
Examples of traumatic neuroma in continuity. Transverse view of a small superficial peroneal nerve neuroma following a motorbike injury (**A**). Transverse view of a median nerve traction neuroma from a distal elbow injury (**B**). Longitudinal view of a large disorganized ulnar neuroma from grenade shard injury to the forerarm (**C**).

**Figure 15 diagnostics-11-00030-f015:**
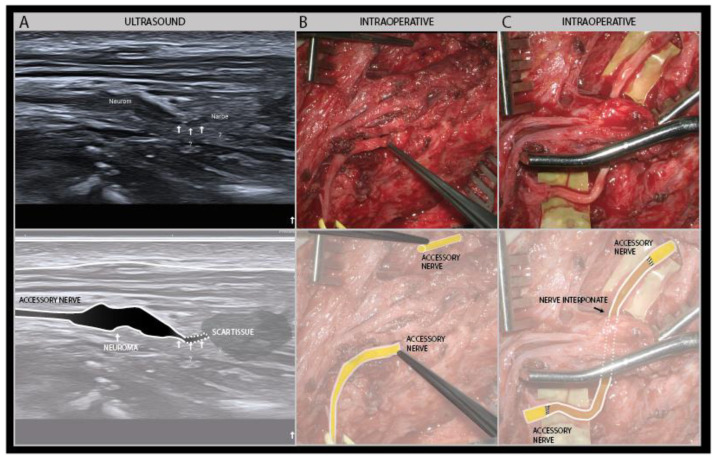
Seventeen-year-old boy with a car accident and injury to the neck due to broken glass shards. There was a complete paresis of the trapezius muscle; and the nerve ultrasound of the accessory nerve showed a neuroma without nerve continuity (high-frequency of 19–24 MHz) (**A**). Intraoperative images show a complete sharp transection injury of the accessory nerve without continuity (**B**). Another intraoperative image after placement of a nerve interponate (**C**) (these intraoperative images were kindly provided by Prof. Dr. G. Antoniadis).

**Table 1 diagnostics-11-00030-t001:** Summary of subcutaneous peripheral nerves at risk during various surgical procedures (for references see text).

ANATOMICAL SITE		
**NECK**		
Nerve	Procedure	Incidence
Accessory nerve	- Local lymph node biopsy or excision	- 3–6 %
**UPPER EXTREMITY**		
Nerve	Procedure	Incidence
Radial nerve	- Surgical approach humeral shaft fracture - Lateral surgical approach to the elbow	- 4–12%
Superficial radial nerve	- Quervain’s disease - Surgical approach distal radius fracture - Distal biceps tendon repair - Venipuncture	- 4% - 20% - 3–17%
Ulnar nerve	- Surgical approach supracondylar humeral fracture - Surgical approach distal humeral fracture - Total elbow replacement	- 2–20% - 5–13% - 5–20%
Median nerve	- Carpal tunnel release surgery - Tendon transfer procedures	- 1.5–15%
Medial antebrachial cutaneous nerve	- Elbow arthroscopy - Brachioplasty (cosmetic surgery)	- 0.2% - 5%
Lateral antebrachial cutaneous nerve	- Distal biceps tendon repair - Venipuncture	- 4–11%
Intercostal brachial nerve	- Dissection of axilla (i.e., lymph node excision)	
**LOWER EXTREMITY**		
Nerve	Procedure	Incidence
Lateral femoral cutaneous nerve	- Total hip replacement - Revision total hip replacement - Iliac crest bone harvesting - Vascular surgery in the femoral triangle	- 2–5% - 10%
Saphenous nerve	- Medial approach meniscectomy - Midline incision knee (terminal saphenous branches) - Great saphenous vein surgery	- 2–9% - 81–100%
Common fibular nerve	- Total knee replacement	- 0.3–9.5%
Superficial fibular nerve	- Lateral approach of the ankle region: - Treatment of distal fibula fracture - Portal placement in ankle arthroscopy	- 2%
Sural nerve	- Repair calcaneal and fibular tendons - Ankle arthroscopy - Distal fibula fracture surgery - Stripping of the small saphenous vein	- 18%

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
