# Peer review of "Nerve Ultrasound in Traumatic and Iatrogenic Peripheral Nerve Injury"

_diagnostics, 2020, doi:10.3390/diagnostics11010030_

Round 1

Reviewer 1 Report

Review is focused to interesting field and can be useful for clinicians.

The preoperative detection of nerve by US is good tools for prevention of iatrogenic injury. The quick and easy localization and detection of injury by US is hgelpfull.

Minor remarks:

Fig. 1A – ad arrow to the nerve

Add to Table 1 – intercostobrachial nerve – dissection of axilla

Radial nerve in elbow region – lateral approcach to the elbow

Table 1

2x Correct spelling of venipuncture is venepuncture

Author Response

Reply to reviewers Diagnostics Traumatic PNS injury ultrasound

Reviewer 1:

Review is focused to interesting field and can be useful for clinicians.

The preoperative detection of nerve by US is good tools for prevention of iatrogenic injury. The quick and easy localization and detection of injury by US is hgelpfull.

Minor remarks:

Fig. 1A – ad arrow to the nerve

We have added an arrow to highlight the nerve in figure 1A

 Add to Table 1 – intercostobrachial nerve – dissection of axilla  Radial nerve in elbow region – lateral approcach to the elbow

Thank you for adding these examples. Table 1 has been edited and the two suggestions mentioned have been incorporated.

2x Correct spelling of venipuncture is venepuncture

Our apologies, but according to our translater service “venipuncture” is a correct English term (although “venepuncture” also seems to be used). We suggest the editor decides on the final spelling desired for this journal.

Reviewer 2 Report

I enjoyed reading this review about nerve ultrasound in traumatic nerve lesions. Wijntjes et al give a comprehensive overview of the topic and the figures are nicely illustrated.

I have only minor concerns:

  • page 2 line 71-78: The sentence is very long and difficult to follow. 
  • page 7 line 213-214: This observation seems obvious, but is this the authors experience, or are there any references supporting it?
  • page 7 line 215: delete "in"
  • page 9 figure 7: the lettering "F" is missing in corresponding part of the figure description
  • page 16 figure 13: description says infraspinatus muscle, while figure itself says supraspinatus muscle
  • table 1: Could you give corresponding references for the data in the table?

Author Response

Reply to reviewers Diagnostics Traumatic PNS injury ultrasound

Reviewer 2:

I enjoyed reading this review about nerve ultrasound in traumatic nerve lesions. Wijntjes et al give a comprehensive overview of the topic and the figures are nicely illustrated.

Thank you.

I have only minor concerns:

page 2 line 71-78: The sentence is very long and difficult to follow.

This sentence has been split in two, and the 2nd sentence has been slighly reworded to make it more clear.

page 7 line 213-214: This observation seems obvious, but is this the authors experience, or are there any references supporting it?

While there are a few reports on sonopalpation in other MSK tissues and 2 studies that mention the option in nerve ultrasound, there are – to our knowledge – no studies that have examined the diagnostic value of what we propose here. We added “In our opion” to the start of this sentence to reflect this lack of evidence.

page 7 line 215: delete "in"

Thank you for noticing; “in” has been deleted

page 9 figure 7: the lettering "F" is missing in corresponding part of the figure description

Apologies for missing this, the letter “F”has now been added to the figure legend

page 16 figure 13: description says infraspinatus muscle, while figure itself says supraspinatus muscle

Thank you for noticing. We adapted the muscle name in the figure to “infraspinatus” (as it should be).

table 1: Could you give corresponding references for the data in the table?

The references for the table were in line 613 where the table is mentioned in the text, but have now also been added to the table legend.

In addition to the reviewers’ comments we made a few other small edits, deleting unnecessary spaces between words and adding an acknowledgement to the surgical images in figure 15.